# In Vivo Trafficking of the Anticancer Drug Tris(8-Quinolinolato) Gallium (III) (KP46) by Gallium-68/67 PET/SPECT Imaging

**DOI:** 10.3390/molecules28207217

**Published:** 2023-10-22

**Authors:** Afnan M. F. Darwesh, Cinzia Imberti, Joanna J. Bartnicka, Fahad Al-Salemee, Julia E. Blower, Alex Rigby, Jayanta Bordoloi, Alex Griffiths, Michelle T. Ma, Philip J. Blower

**Affiliations:** 1College London, School of Biomedical Engineering and Imaging Sciences, St. Thomas’ Hospital, London SE1 7EH, UKfahad.al-salemee@nhs.net (F.A.-S.); alex.rigby@kcl.ac.uk (A.R.); michelle.ma@kcl.ac.uk (M.T.M.); 2Department of Radiologic Sciences, Faculty of Applied Medical Sciences, King Abdulaziz University, Jeddah 21589, Saudi Arabia; 3London Metallomics Facility, King’s College London, London SE1 9NH, UK

**Keywords:** gallium, KP46, 8-hydroxyquinoline, radionuclide imaging, PET imaging, SPECT imaging, cancer

## Abstract

KP46 (tris(hydroxyquinolinato)gallium(III)) is an experimental, orally administered anticancer drug. Its absorption, delivery to tumours, and mode of action are poorly understood. We aimed to gain insight into these issues using gallium-67 and gallium-68 as radiotracers with SPECT and PET imaging in mice. [^67^Ga]KP46 and [^68^Ga]KP46, compared with [^68^Ga]gallium acetate, were used for log*P* measurements, in vitro cell uptake studies in A375 melanoma cells, and in vivo imaging in mice bearing A375 tumour xenografts up to 48 h after intravenous (tracer level) and oral (tracer and bulk) administration. ^68^Ga was more efficiently accumulated in A375 cells in vitro when presented as [^68^Ga]KP46 than as [^68^Ga]gallium acetate, but the reverse was observed when intravenously administered in vivo. After oral administration of [^68/67^Ga]KP46, absorption of ^68^Ga and ^67^Ga from the GI tract and delivery to tumours were poor, with the majority excreted in faeces. By 48 h, low but measurable amounts were accumulated in tumours. The distribution in tissues of absorbed radiogallium and octanol extraction of tissues suggested trafficking as free gallium rather than as KP46. We conclude that KP46 likely acts as a slow releaser of gallium ions which are inefficiently absorbed from the GI tract and trafficked to tissues, including tumour and bone.

## 1. Introduction

Following the discovery that the gamma-emitting radionuclide gallium-67 (^67^Ga) is taken up specifically in tumours (particularly lymphoma) [1], ^67^Ga has become valuable in nuclear medicine as an imaging agent for diagnosing lymphoma, inflammation, and infection using scintigraphy or single-photon emission computed tomography (SPECT) [2,3]. It is also being evaluated as a therapeutic radionuclide by virtue of its Auger–Meitner electron emissions [2,3,4]. The positron-emitting isotope ^68^Ga has also found widespread application in positron emission tomography (PET) [2]. The discovery also sparked interest in the potential of nonradioactive gallium as the basis of drugs for treatment of cancer and other disorders. The development of gallium-based drugs, primarily as anticancer agents, began with simple salts including gallium(III) nitrate (marketed as Ganite^TM^) [5], gallium(III) chloride [6], and gallium(III) citrate [7]. There is little chemical basis to distinguish between these forms from a pharmacology perspective—all contain hydrated and hydrolysed Ga^3+^ ions, and despite its formal description as gallium nitrate, the approved formulation of Ganite^TM^ contains citrate. Ganite^TM^ showed particular promise in clinical trials in patients with non-Hodgkin’s lymphoma [8] and bladder cancer [7,9] and is approved for treatment of malignancy-related hypercalcemia [9,10,11]. It is administered as an intravenous infusion. Gallium citrate (Panaecin^TM^) [12] is being evaluated in clinical trials as an inhaled formulation for the treatment of a variety of lung infections. Gallium chloride has been evaluated clinically and preclinically for treatment of various cancers [13,14,15]. A range of mechanisms of anticancer and other actions have been suggested, but most mechanistic investigations focus on the downstream consequences of interference with iron transport and metabolism related to the chemical similarity in ligand-binding characteristics between Ga(III) and Fe(III) [4,7,15,16,17,18].

These promising examples of therapy with gallium salts involved primarily intravenous administration. Oral administration would make the drugs more acceptable and would be expected to reduce toxic side effects. However, bioavailability of gallium in the blood after oral administration of gallium nitrate [11], citrate [19], and chloride [11,13,20] in animal models was very poor. The search for orally administered forms of gallium led to the evaluation of second-generation compounds: gallium tris(8-hydroxyquinolinate) (known as KP46 or AP002, Figure 1) and gallium tris(maltolate), both designed as more lipophilic compounds in the expectation of improved absorption [18]. In clinical trials, KP46 showed promise against renal cancer [21]; furthermore, as AP-002, it is currently in phase I–II clinical trials (national clinical trial identifier (NCT) 04143789) for patients with breast, lung, and prostate cancer and bone metastases. However, in vivo preclinical data suggest that it, too, has poor bioavailability [21,22], possibly related to its very low water solubility [23,24]. Ga-maltolate has higher water solubility than KP46, and its oral administration leads to levels of gallium in serum, mainly bound to transferrin, comparable to those achieved during Ganite^TM^ infusion. Gallium maltolate is currently the subject of a clinical trial in glioblastoma (NCT04319276). 

Despite numerous studies both in vitro in cancer cell lines and in vivo in rodent models, and despite the compilation of detailed and critical reviews [25], the absorption, speciation, pharmacokinetics, and trafficking of gallium after administration of KP46 and gallium maltolate remain poorly understood. KP46 shows cytotoxic activity in some cell lines when added directly to cultured cells [16,18,19,25,26,27,28,29,30], but since the speciation of gallium in vivo en route to the tumours is not fully elucidated, it is unclear whether direct treatment of cultured cells with KP46 is relevant to the in vivo and clinical context. Investigations of transchelation and binding of gallium to serum proteins, particularly transferrin and albumin, have given conflicting results: some works suggest transchelation of gallium from KP46 to transferrin and other data suggest hydrophobic association of the intact complex with apo-transferrin and albumin [31,32,33].

**Figure 1 molecules-28-07217-f001:**
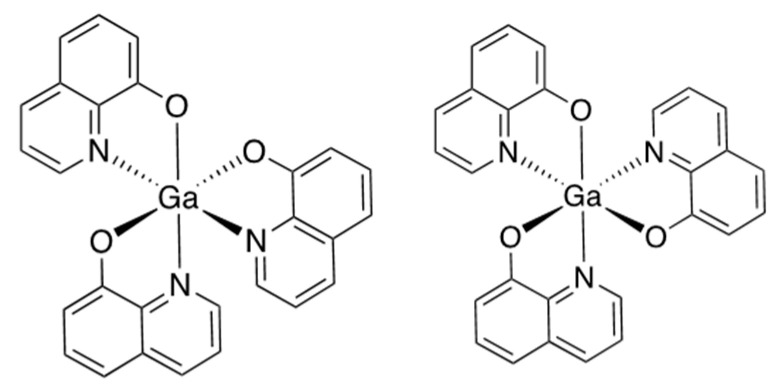
Structure of tris(8-quinolinolato)gallium (III) (KP46) [34]. **Left**: *mer*-isomer; **right**: *fac*-isomer.

Thus, despite clinical trials being in progress and some being completed, understanding of the absorption, speciation, and pharmacokinetics of KP46 and gallium maltolate in vivo and of the trafficking of gallium to tumours after their administration remains poor and must be improved if enhanced design and delivery of gallium drugs is to be achieved. It was briefly pointed out [25] that radionuclides such as ^68^Ga, mentioned above, could, in principle, be used to illuminate gallium trafficking in this context, but such studies have not been reported despite increased interest recently in the use of radionuclide imaging to study the biology of trace metals and metallodrugs [35,36,37]. Here, we describe for the first time the use of radionuclides ^67^Ga and ^68^Ga to help understand the speciation and trafficking of gallium administered in the form of KP46 in vitro and in vivo in a tumour-bearing mouse model using PET and SPECT imaging. The tumour cell line used here is the A375 human melanoma cell line, chosen because KP46 has been shown to be active against human melanomas [28] and because in our laboratory, A375 tumours in mice have shown significant avidity for i.v. injected ^68^Ga in the form of gallium chloride or acetate [38,39].

## 2. Results

### 2.1. Synthesis and Radiolabelling of KP46

Synthesis and characterisation of KP46 [22] and its radiolabelled forms were carried out as described in the Materials and Methods section below and in the Appendix A. Detailed spectroscopic and chromatographic/radiochromatographic studies were carried out (also detailed in Appendix A) to ensure that the speciation and physicochemical characteristics of the various radiolabelled forms were homogeneous with those of the bulk samples of nonradioactive KP46 used in the biological experiments.

No-carrier-added (“tracer level”) [^68^Ga]KP46 and [^67^Ga]KP46 were produced by mixing 8-hydroxyquinoline with, respectively, ^68^Ga- or ^67^Ga-chloride in acetate buffer, resulting in efficient chelation, to form a radioactive lipophilic complex. The radioactive product was purified by elution with ethanol (recovering >90% of activity) from a disposable reverse-phase cartridge column and diluted with water. Radiochemical purity of the product was >98%, as determined by radio-iTLC (instant thin layer chromatography) with a mobile phase of CHCl_3_:CH_3_OH (95:5%, *v*/*v*) with Rf = 0.8–1.0 (using the same iTLC conditions, [^68/67^Ga]Ga-acetate/chloride showed Rf = 0, see Appendix A). iTLC of the nonradioactive bulk samples of KP46, detected by the visible and UV-active yellow spot, showed the same Rf value of 0.8–1.0. [^67^Ga]KP46 behaved similarly (Appendix A). The radiochemical purity was also confirmed by reverse-phase HPLC, in which radioactivity ([^67^Ga]KP46) coeluted with KP46, with an elution time of 16.6 min (gamma detection) and 16.1 min (UV); the difference matches the delay time between UV radioactivity and UV detectors, as shown in Appendix A.

### 2.2. Distribution/Partition Coefficients (logD and logP) of [^68^Ga]KP46

As expected, [^68^Ga]KP46 proved to be highly lipophilic, with log*D*_(octanol/PBS)_ and log*P*_(octanol/water)_ values of 1.90 ± 0.19 and 2.33 ± 0.28, respectively. In contrast, [^68^Ga]gallium acetate proved, as expected, to be highly hydrophilic, with log*D* and log*P* values of −3.00 ± 0.3 and −3.20 ± 0.48, respectively (Appendix A).

### 2.3. In Vitro [^68/67^Ga]KP46 Binding to Serum Proteins

Binding to apo-transferrin (apo-Tf) and human serum albumin (HSA) was probed by size exclusion chromatography using small cartridge columns, from which both apo-Tf and HSA (controls, detected by UV absorbance at 280 nm) eluted in fractions 3–5 (1.5–2.5 mL, Appendix A). Incubation of [^68/67^Ga]gallium acetate with apo-Tf led to elution of radioactivity exclusively in fractions 3–5, indicative of complete binding of gallium to apo-Tf (Figure 2). In contrast, [^68/67^Ga]KP46, after similar incubation with or without apo-Tf, eluted in fractions 12–23, with almost no radioactivity eluted in fractions 3–5 (Figure 2).This indicates that the radiogallium largely remains as small molecules with little tendency to undergo transchelation to apo-Tf and that intact [^68/67^Ga]KP46 does not strongly associate with apo-Tf even after a 48 h incubation. When incubated with HSA, [^68^Ga]gallium acetate showed partial binding to protein (eluting in fractions 3–5 as well as in later fractions (5–17), see Figure 2). Incubating [^68^Ga]KP46 under these conditions led to about 16% of radioactivity coeluting with HSA, with the remainder eluting in fractions 12–23 (Figure 2). Thus, both ionic gallium and KP46 showed evidence of partial binding of gallium to HSA. An increase in column residual activity was noticed with time (see Figure 2), particularly in the apo-Tf experiments. This might be related to the formation with time of insoluble gallium hydroxide that was retained in the column. 

### 2.4. Cellular Uptake of [^68^Ga]KP46

Upon incubation in vitro with cultured A375 cells in DMEM medium containing foetal bovine serum (FBS), both [^68^Ga]KP46 and [^68^Ga]Ga-acetate exhibited accumulation in cells after 1 h (that is, both tracers led to a higher concentration of radiogallium within cells than in the medium). When corrected for nonspecific binding of radioactivity to the vessel walls in the absence of cells, [^68^Ga]KP46 showed significantly higher cellular uptake (2.55% ± 1%) than [^68^Ga]gallium acetate (1.0% ± 0.08%) (*p* < 0.01) in A375 cells (Figure 3A). These data lead to an estimation of intracellular-to-extracellular concentration ratios of 14 ± 6 for [^68^Ga]KP46 and 6 ± 4 for [^68^Ga]Ga-acetate (Figure 3B).

### 2.5. PET Imaging and Biodistribution of Intravenously Injected [^68^Ga]KP46 and [^68^Ga]Ga-Acetate in Mice

The biodistribution of [^68^Ga]KP46 and [^68^Ga]Ga-acetate, each at tracer (no-carrier-added) concentrations, was determined in NSG mice bearing A375 human melanoma xenografts over a period of four hours (a limit imposed by the 68 min half-life of ^68^Ga) after intravenous administration, by PET/CT imaging. Injection of [^68^Ga]gallium acetate led to the visible uptake of ^68^Ga in the tumour by four hours, rising to ca. 6% ID/g. This was accompanied by increasing uptake in the joints and urinary bladder during this period (Figure 4A)—a biodistribution pattern typical of ^67/68^Ga salts [38]. Injection of [^68^Ga]KP46, on the other hand, led to no visible tumour accumulation of ^68^Ga during the four-hour PET/CT scan post-i.v. injection; instead, radioactivity was taken up largely in the liver, with subsequent translocation into the intestines, and in the myocardium (Figure 4B). The ex vivo biodistribution of both tracers at four hours confirmed the PET observations and showed significantly higher tumour uptake after injection of [^68^Ga]gallium acetate (6.1 ± 2.4%ID/g) than [^68^Ga]KP46 (2.6 ± 0.9%ID/g) (Figure 4C and Appendix A). Additional ex vivo biodistribution data for [^68^Ga]KP46 at 2 h p.i. are shown in Appendix A.

### 2.6. Four-Hour PET Imaging and Biodistribution of Orally Administered “Tracer” and “Bulk” [^68^Ga]KP46 in Mice

In the imaging protocol we developed, imaging prior to 4 h postadministration was not performed to avoid general anaesthesia in the hours after administration, which adversely affects gastric emptying and translocation through the gut. Both tracer (no-carrier-added, group C) and bulk, pharmacologically relevant doses of [^68^Ga]KP46 (group D) were investigated. After the oral administration of [^68^Ga]KP46 as a tracer, PET/CT images showed an uptake pattern consistent with rapid transit from stomach to small and large intestine (Figure 5). Despite the expectation that the liver would be the main initial repository (delivered via the portal circulation) for radioactivity absorbed in the intestines, very little liver activity, or indeed activity in any other tissues, was observed. Ex vivo biodistribution results (Figure 5 and Appendix A) supported these PET/CT results, showing uptake in the large intestine (61.7 ± 21.5%ID/g) and small intestine (12.8 ± 4.1%ID/g) and only very low activity in liver and urine.

After the oral administration of [^68^Ga]KP46 combined with KP46 (“bulk”, group D), the PET/CT images showed an uptake pattern similar to that of the tracer-level administration, with high ^68^Ga activity in the large intestine (56.5 ± 21.4%ID/g), small intestine (18.4 ± 5%ID/g), and stomach (11.9 ± 8.1%ID/g). Other tissues showed little ^68^Ga uptake. Negligible activity was delivered to the tumour (0.5 ± 0.3%ID/g). Because of the bulk pharmacological dose administration, analysis of ^69^Ga in the tissues by ICP-MS was possible. This analysis showed a similar biodistribution pattern to that of the ^68^Ga, indicating that ^68^Ga was reliably tracking the bulk nonradioactive gallium (Appendix A). ^69^Ga was seen mostly in the large intestine (23.7 ± 9.2%ID/g) and a negligible amount of ^69^Ga was absorbed from the gut and delivered to the tumour (0.17 ± 0.1%ID/g). 

To address the question of whether the [^68^Ga]KP46 was chemically intact after 4 h within the digestive tract and in blood serum and urine, the lipophilicity of ^68^Ga in these tissues from mice in group C was measured by octanol extraction (Figure 6). This approach exploits the knowledge that intact KP46 is efficiently extracted into octanol, whereas ^68^Ga salts (e.g., acetate) are retained in the aqueous phase. ^68^Ga in the serum and urine samples displayed hydrophilic behaviour (98.6% ± 1.5% and 98.2% ± 1.7%, respectively, was found in the aqueous phase). Most of the activity in the stomach (82.9% ± 10.7%), small intestine (80.7% ± 19.4%) and large intestine (83.7% ± 8.8%) contents was present in solids and not extracted into either phase (Figure 6A). After removal of these solids, solvent extraction of the liquid phase showed the majority of radioactivity in the aqueous phase (76.8% ± 8.7% for stomach, 95% ± 2.5% for small intestine, and 95.5% ± 4.2% for large intestine) (Figure 6B). This suggests the ^68^Ga had been to a significant extent converted to a hydrophilic form such as ionic gallium. 

### 2.7. Twenty-Four- and Forty-Eight-Hour PET Imaging and Biodistribution of Orally Administered “Tracer” and “Bulk” [^67^Ga]KP46 in Mice

Because the time scale of in vivo experiments with ^68^Ga was limited by its short half-life (68 min), investigations of the biodistribution of orally administered KP46 were repeated using the longer-lived (78.3 h) ^67^Ga gamma-emitting radionuclide ^67^Ga. SPECT imaging with ^67^Ga enabled the in vivo fate of gallium to be tracked for 4, 24, and 48 h, both at tracer-only level (group E) and bulk pharmacological level (group F). In both groups, SPECT/CT images at the four-hour time point were consistent with the four-hour [^68^Ga]KP46 PET study described above, with radioactivity largely confined to the stomach and intestines (Appendix A). At 24 h, however, significant differences emerged between the “tracer level” group and the “bulk level” group. In the “tracer level” group (E), SPECT/CT imaging showed little radioactivity in any organs, and the vast majority of radioactivity had been excreted in faeces (as evident from both the SPECT scans and radioactivity measurements of the cage contents as well as animal tissues). In the “bulk level” group (F), on the other hand, faecal excretion was significantly delayed, and a much higher fraction of radioactivity remained in the body. This delay allowed time for absorption of some ^67^Ga from the gut and translocation to other tissues; although substantial radioactivity remained in the intestines, measurable amounts appeared in the liver, joints, and tumour (Figure 7). Because of low levels of ^67^Ga retained within the “tracer level” group E animals, imaging was not continued, and the animals were culled after imaging at 24 h. Ex vivo biodistribution of radioactivity in these animals (Appendix A) confirmed the SPECT/CT results and showed that a considerable amount of activity was in faeces (3.0 ± 2.9%ID/g).

Mice in the “bulk level” group were imaged again 48 h postadministration of the [^67^Ga]KP46. By this time, although much of the radioactivity had been excreted in faeces, the images showed visible uptake in tumour, liver, and joints (Figure 8A). After imaging, these animals were culled and ex vivo biodistribution was determined (Figure 8B, Appendix A), providing data that were consistent with the 48 h SPECT imaging: there was significant uptake in bone (2.55 ± 1.82%ID/g), liver (1.34 ± 1.01%ID/g), and tumour (1.57 ± 0.48%ID/g). 

Serum and urine from the “tracer level” group (E) were subjected to octanol extraction 24 h postadministration to determine the fraction of ^67^Ga that may have survived in the form of [^67^Ga]KP46 after absorption from the gut. Less than 6% of serum radioactivity, and less than 3% of urine activity, was extracted into the octanol phase, indicating that very little of the absorbed ^67^Ga was in the form of intact [^67^Ga]KP46 by 24 h. 

Tissues harvested from the “bulk level” group (group F) after imaging and ex vivo radioactivity measurement at 48 h were digested and analysed by ICP-MS to determine ^69^Ga levels. The analysis showed the highest ^69^Ga levels in the intestines and faeces but also detectable levels in the tumour and liver (Appendix A). Although not all tissues were analysed due to failure to completely digest into solution, the lack of major discrepancy between ^69^Ga levels measured by ICP-MS and ^67^Ga levels measured by gamma counting suggests that ^67^Ga acts as a reliable tracer for overall gallium distribution when [^67^Ga]KP46 is coadministered orally with KP46.

## 3. Discussion

A comparison of the chromatographic and radiochromatographic properties (iTLC and HPLC) of ^68^Ga- and ^67^Ga-labelled KP46 indicates that the labelled species can be expected to be a reliable tracer for the bulk KP46 drug. The in vivo experiments, where biodistributions of the radionuclides (using radioactivity measurements) and the nonradioactive gallium-69 were compared, indicated that the radionuclides were also reliable tracers for gallium administered as the KP46 drug in vivo. The additional detail of gallium distribution provided by radionuclide imaging, over and above the direct analysis of nonradioactive gallium in tissue samples, is therefore potentially highly informative. For example, using radionuclide imaging, the distribution of gallium can be studied at a range of different time points without adding to the number of animals, and additional insight is accessible into the biodistribution in tissues that would not typically be sampled ex vivo or that would be hard to sample ex vivo. For example, the SPECT images show that uptake in the bones is primarily in the joints. Thus, we conclude that ^68^Ga PET imaging and ^67^Ga SPECT imaging represent useful, and hitherto unexploited, tools with which to study the pharmacokinetics of gallium-based drugs. Subject to satisfactory dosimetric evaluation and regulatory approvals, this utility potentially extends to human studies, since these radionuclides are widely used in nuclear medicine, and human PET and SPECT scanners are widely available in hospitals.

It is acknowledged that the ability of a drug compound to be absorbed across the gastrointestinal barrier can usefully be predicted by measurement of its lipophilicity, e.g., by use of octanol extraction to determine the distribution/partition coefficient [5,23,39]. KP46 was first introduced as a lipophilic form in which to deliver gallium orally, and measurement of the octanol–water partition coefficient of the bulk drug spectrophotometrically, confirming its lipophilicity (log*P*_(octanol/water)_ = 0.88), is described in the literature [23,25]. In the present work, the octanol–water partition coefficient was determined using radioactivity measurements at the tracer level, again confirming its lipophilicity but showing a much higher log*P* value (2.33 ± 0.28). We suggest that the origin of the difference lies in the different methods and conditions: bulk KP46 contains Ga^3+^ and 8-hydroxyquinolinate in a 1:3 ratio and is subject to dissociative equilibria in solution to form complexes with a higher Ga:8HQ ratio and which would consequently be positively charged and less lipophilic than the 1:3 complex. The apparent log*P* value determined spectrophotometrically using the bulk drug is therefore likely to be an underestimate of the true value for the 1:3 complex itself. The ^68^Ga measurement, on the other hand, was performed on a solution containing tracer (<nanomolar) levels of ^68^Ga but a large excess—more than a millionfold—of 8-hydroxyquinoline (a necessary consequence of the radiochemical synthesis). This excess is likely to suppress dissociation, ensuring that virtually all of the ^68^Ga is in the form of the 1:3 complex and hence more efficiently extracted into octanol, giving a higher log*P* that more closely reflects the lipophilicity of the 1:3 complex. This dissociative equilibrium is also suggested by other aspects of the physicochemical characterisation of KP46, including HPLC and mass spectrometry. It is likely to be an important consideration in understanding the absorption of the drug and its interaction with serum proteins. Reverse-phase gradient HPLC of KP46 with UV detection (Appendix A) shows an unusual line shape suggestive of on-column dissociative equilibria: an unusually broad peak from 14 to 16 min merging into a sharp peak at 16.2 min. Mass spectrometry of the late (16.2 min), sharp part of this peak suggests that it represents primarily the 1:3 complex, (M + H^+^ 503.77 for the ^71^Ga isotope). By contrast, mass spectrometry of the earlier broad peak (14.1 min) shows no peak representing the 1:3 complex; instead, it shows predominantly uncomplexed 8HQ (M + H^+^ 146). Other minor peaks observed in mass spectra were consistent with the presence of complexes with lower Ga:8HQ ratios (e.g., 1:2, 2:5, see Appendix A). Although MS alone cannot determine whether these species were present in solution rather than being formed during the electrospray ionisation process, its combination with the HPLC peak shape and the partition coefficient data strongly suggests that dissociative equilibria are important.

Radiogallium-labelled KP46 (^68^Ga initially, for short incubations, and ^67^Ga when extension to longer time points was required) proved useful in understanding interactions of the drug with serum proteins. These become relevant if KP46 is administered intravenously or is absorbed intact from the GI tract. Apo-transferrin (apo-Tf) and human serum albumin (HSA) are likely candidate proteins for binding to KP46 or gallium released from it: apo-Tf because, as an iron(III)-binding protein, it is known to bind with high affinity to Ga^3+^ in the presence of bicarbonate ions [15,33,40,41], and HSA because of its high abundance and known possession of binding sites for metals and lipophilic molecules.

In experiments with apo-Tf, while ionic Ga^3+^ (as its acetate salt) showed the expected almost complete binding to apo-Tf in the presence of bicarbonate, KP46 by contrast showed no evidence of gallium binding to apo-Tf. This shows that neither transchelation of gallium nor hydrophobic binding of the intact KP46 complex occurred under the conditions of the experiment. The elution time of the radioactivity matched that of KP46 controls (where protein was absent), suggesting that KP46 remains intact during the incubation. This result conflicts with previous studies of KP46 binding to apo-Tf in which extensive binding was observed using bulk nonradioactive KP46 [32,33,42] either with or without dissociation of the KP46 complex [31,32]. Again, we suggest that the differing methods and conditions may account for the apparent conflict. In the “bulk drug” approaches reported in the literature, the KP46 was presented to the protein as a stoichiometric compound with a Ga:8HQ ratio of 1:3, whereas in the radiotracer experiments, gallium was present at a very low concentration with 8HQ in very large excess. In the small volume of serum used, this excess may be sufficient to overcome the greater affinity of gallium for apo-Tf than for 8HQ and shift the equilibrium away from transchelation of gallium to apo-Tf. This argument is consistent with the view that, if KP46 is absorbed intact from the digestive tract into blood, dissociative equilibria become important in determining its pharmacokinetics and biodistribution. 

Upon incubation with HSA, both [^68/67^Ga]KP46 and [^68/67^Ga]gallium acetate showed only weak protein association, consistent with the literature [33]. In the case of [^68/67^Ga]KP46, the HSA-binding data alone provide no indication of whether the binding is due to transchelation or a hydrophobic interaction between HSA and intact KP46; however, in light of the apo-Tf binding data, under these conditions, where dissociation of KP46 is suppressed by the presence of excess 8HQ, transchelation is unlikely [33,42]. Indeed, studies in the literature using NMR and XANES suggest a hydrophobic binding mechanism for association of bulk KP46 with HSA in vitro [32,43,44,45].

In vitro studies using the A375 cell line were conducted to compare the cellular uptake of [^68^Ga]KP46 with that of ionic gallium. No-carrier-added [^68^Ga]KP46 was used without addition of nonradioactive KP46. With the levels of activity used in these experiments, ^68^Ga concentration added to the media under these conditions was in the pM–nM range. Both [^68^Ga]KP46 and unchelated ^68^Ga ([^68^Ga]gallium acetate) were accumulated in cells, leading to higher radiogallium concentration within cells than in the medium. The fact that [^68^Ga]KP46 showed a significantly higher uptake (*p* < 0.01) compared to [^68^Ga]Ga-acetate is attributable to the lipophilicity of [^68^Ga]KP46, which allows cell entry by passive diffusion. This shows that transferrin-dependent or other specific transport mechanisms are not necessary for KP46 to deliver gallium into cells. The 8HQ complexes of ^68^Ga and ^67^Ga are known to be able to transport radiogallium nonspecifically into a variety of cells by virtue of their lipophilicity [2,4,46]. Indeed, this has been used as a method of radiolabelling cells for tracking their migration in vivo [46]. Unchelated gallium is also known to be taken up in cells via transferrin-mediated and possibly non-transferrin-mediated mechanisms [4,16,17,41,44]. In the media used here, which contain foetal bovine serum, bovine transferrin is present; the sluggish uptake of gallium acetate in cells can be accounted for by noting that although gallium binds to bovine transferrin with similar efficiency as it does to human transferrin, human transferrin receptor has relatively poor affinity for bovine transferrin. The presence of bovine transferrin, therefore, could have an inhibitory effect on the uptake of gallium ions into human cells by diminishing the availability of gallium for transport into cells via non-transferrin-mediated routes [47,48,49,50]. This inhibition would not affect the KP46-mediated transport into cells. The experiments show that if KP46 is absorbed intact from the digestive tract into blood and survives intact in blood en route to the tumour, it could deliver gallium into tumour cells.

A comparison of [^68^Ga]gallium acetate and [^68^Ga]KP46 biodistribution after intravenous injection at the tracer level using PET imaging and ex vivo tissue counting showed clear differences between the two. In the case of [^68^Ga]gallium acetate, steady clearance of ^68^Ga from the blood pool was observed, leading by 4 h to significant (visible above background in scans) uptake in the tumour and joints and urinary excretion. This is consistent with previous ^68^Ga and ^67^Ga imaging studies and with the accepted transferrin-dependent distribution pathways [2,4,17,38,44]. In contrast, i.v. administration of [^68^Ga]KP46 led to rapid blood clearance of ^68^Ga and uptake in the liver and myocardium within 30 min, with later translocation of radioactivity from the liver to intestines. This indicates that the [^68^Ga]KP46 remains intact without dissociation in blood for the brief period required to prevent ^68^Ga from distributing via transferrin binding. This is consistent with the in vitro serum behaviour of [^68^Ga]KP46 at tracer level, where the presence of excess 8HQ suppresses transchelation to transferrin. It must be noted that these conditions do not mimic those that result from the typical oral administration route of KP46 in clinical trials; instead, they indicate how KP46 might be expected to behave if absorbed intact from the digestive tract. Notably, the tumour is not among the organs receiving a significant fraction of ^68^Ga after intravenous administration of [^68^Ga]KP46, despite the demonstrated ability of [^68^Ga]KP46 to deliver ^68^Ga to tumour cells in vitro. We assume this difference between in vitro and in vivo behaviour is due to the rapid absorption of tracer by the liver in vivo, removing ^68^Ga from circulation too quickly to allow significant delivery to the tumour. 

PET imaging 4 h after oral administration of [^68^Ga]KP46, both at tracer level and as part of a bulk dose, showed that radioactivity was largely confined to the digestive tract, with very little absorption; the little activity outside the gut was present in the liver and urine, with no significant tumour uptake. This is indicative of very poor absorption of ^68^Ga by 4 h. 

To determine whether more significant absorption occurs over a longer time period, similar studies were conducted with gallium-67. After oral administration of [^67^Ga]KP46 at a tracer level or combined with bulk KP46, SPECT imaging results after 4 h were consistent with those obtained with ^68^Ga: most of the activity remained in the gut, with minimal absorption and trafficking to tissues (Appendix A). However, at the 24 h time point, the ^67^Ga distribution in the “tracer” and “bulk” groups diverged. [^67^Ga]KP46 administered at the tracer level showed little visible uptake in any organ (including tumour) in SPECT/CT images and had been largely excreted via faeces. Ex vivo tissue counting revealed small but measurable activity in tissues outside the digestive tract; the highest activity other than the digestive tract and faeces was found in the liver and tumour, followed by the skeleton. In contrast, the presence of a therapeutic bulk dose of KP46 delayed faecal excretion, and at 24 h, while much of the radioactivity still remained in the gut, SPECT scans showed translocation of a small but measurable fraction of ^67^Ga to other organs, including tumour. At this point, the biodistribution of ^67^Ga in tissues outside the digestive tract, particularly the activity in the joints, resembled that of i.v.-administered gallium acetate rather than that of i.v.-administered KP46. By 48 h, this distribution was further accentuated: most gut radioactivity had been eliminated by faecal excretion and by further absorption, leading to higher radioactivity concentration in tissues, including tumour, compared to 24 h. The highest activity was found in bone (mainly in joints as identified in SPECT scans), followed by tumour and then liver. Again, this is reminiscent of the biodistribution shown by gallium acetate rather than KP46, although the liver-to-tumour and liver-to-bone ratios were slightly higher than when ^68^Ga acetate was administered by i.v. injection. This difference should be viewed in light of the different delivery routes: i.v.-administered tracer reaches tissues/tumour after first passing through the pulmonary circulation, whereas orally administered tracer first passes through the portal circulation to the liver before redistribution to the pulmonary circulation and then other tissues. Hence, the liver has the greatest opportunity to capture the orally administered tracer. Attempts to determine the speciation of ^68^Ga at 4 h post-oral administration by octanol extraction showed no evidence of a significant lipophilic form of ^68/67^Ga in any nondigestive tissue including tumour; this is consistent with the conclusion that once absorbed from the gut, the gallium is handled similarly to gallium acetate rather than intact KP46. A similar analysis of the gut contents was attempted, but the majority of radioactivity was insoluble in both the octanol and aqueous layers, and amongst the soluble fraction, less than 20% was octanol-extractable, indicating that extensive hydrolysis and dissociation of KP46 had occurred in the stomach and intestines even by 4 h. Thus, it remains unclear whether the radioactive species absorbed from the gut is KP46 or ionic gallium. However, the octanol extraction of nondigestive tract tissues and the biodistribution of absorbed radiogallium strongly suggest that, at least from the point of passing through the liver, gallium is transported as if administered intravenously in the form of gallium acetate and is, hence, delivered to tissues via transferrin-dependent pathways. The inefficient absorption from the gut ensures that the gallium concentration in the circulation remains low and is unlikely, therefore, to overload apo-transferrin capacity for gallium binding. 

The radiotracer and imaging data presented here complement previous studies using non-radioactive-gallium-based drugs and provide additional insight. In this study, we did not investigate the absorption of orally administered radiogallium salts. Our conjecture is that KP46 (which itself is poorly absorbed at best) dissociates in the digestive tract, releasing ionic gallium, which is then also poorly absorbed and largely excreted faecally. This is consistent with previous studies of absorption and excretion of orally administered gallium salts in mice. In a study of orally administered [^67^Ga]gallium citrate, between 87 and 100% of ^67^Ga was excreted in the faeces by 72 h [19,51]. Similarly, in mice receiving simultaneous single oral administration of bulk gallium chloride (100 nmol) and 0.037 MBq of ^67^Ga citrate, only 0.3 ± 0.08% of the original measured activity remained in the mice after five days [52]. The present study was primarily concerned with the trafficking of KP46, and the results are consistent with a previous study [22] in healthy mice given prolonged (2 weeks) daily oral doses of KP46. This study reported poor absorption, with the highest accumulation of gallium in bone (7.02 ± 3.14 µg/g, 0.16 ± 0.07%ID/g), followed by liver (3.55 ± 2.10 µg/g, 0.08 ± 0.04%ID/g) (there were no tumours in the study). This biodistribution, like that which we observed with [^67^Ga]KP46, indicates that once absorbed and in circulation, the biodistribution pathway matches the known behaviour of unchelated gallium rather than intact KP46. In particular, the biodistribution is not dominated by the high liver or myocardial uptake characteristic of intravenously injected KP46. None of the data presented here suggest that KP46 offers a significant advantage over water-soluble unchelated gallium in terms of absorption and bioavailability. Data are consistent with a model in which KP46 serves as a slow-release form of gallium, releasing ionic gallium in the gut, which is then absorbed rather inefficiently (most is excreted) and which then behaves like free gallium in the blood, with much of it quickly bound to the iron-binding sites of transferrin. However, the possibility cannot be excluded that KP46 is absorbed into the blood and quickly delivered to the liver via the portal circulation, where it is metabolised and rereleased into circulation as free gallium. There is no evidence that orally administered KP46 is delivered intact to tumours in vivo.

The very low delivery of gallium to tumours, shown by the present and previous studies with orally administered KP46, can be interpreted as an indication of the potency of gallium when it does reach the tumour in very small quantities in clinical trials that show effectiveness. Alternatively, its effect on tumours may not require delivery of Ga to tumours; for example, interference with delivery of essential iron to tumours may be implicated. However, all evidence shown here points to KP46 acting as a prodrug, releasing free gallium which becomes the absorbed and active species. The corollary to this mechanism, however, is that KP46 must also release 8-hydroxyquinoline in this process. The absorption, trafficking, and cytotoxicity of 8HQ do not appear to have been considered as a potential contributor to the anticancer properties of KP46. 8HQ has shown cytotoxic effect against multiple cancer cell lines [53,54], and hence, the idea that KP46 could act as a prodrug not only for gallium ions but also for 8HQ should be considered [53,55].

## 4. Conclusions

In summary, we can conclude that the use of radioactive gallium (^68^Ga and ^67^Ga), particularly in conjunction with PET and SPECT imaging, respectively, has a significant contribution to make in elucidating the absorption and trafficking of gallium drugs. Indeed, subject to dosimetry and regulatory evaluation, there are no significant barriers to extending imaging experiments from mice to humans as part of future clinical trials, since clinically usable ^68^Ga and ^67^Ga and clinical PET and SPECT scanners are widely available. Using ^68^Ga and ^67^Ga as tracers, we showed that when KP46 enters the circulation intact, the liver and myocardium are the primary targets, whereas ionic gallium primarily targets the joints and tumour. The biodistribution of gallium following oral administration of KP46 shows extremely inefficient absorption of gallium, and the biodistribution of the small fraction that is absorbed matches that of i.v.-administered ionic gallium and not that of i.v.-administered KP46. Hence, it can be concluded that the gallium arriving at the tumour is not in the form of KP46. In vitro protein-binding studies with radiogallium combined with octanol extraction studies of the stomach and intestine contents suggest that the dissociation occurs not in the blood but in the gut.

## 5. Materials and Methods

### 5.1. General

Reagents and consumables were purchased from Fisher Scientific/Chemical, Crawford Scientific, or Sigma Aldrich unless specified otherwise. Athymic nude *nu*/*nu* mice and NSG mice were purchased from Charles River UK Ltd. (Margate, UK). A375 human melanoma cells were purchased from the American Type Culture Collection (ATCC) and were cultured and stocked in our laboratory. ^68^Ga was obtained from a ^68^Ge/^68^Ga generator (Eckert & Ziegler, Braunschweig, Germany) by eluting with 5 mL of 0.1 M ultrapure hydrochloric acid and collected in 5 × 1 mL fractions. [^67^Ga]Ga-citrate was purchased from Mallinckrodt, The Netherlands. iTLC was carried out using Agilent’s silica-gel-impregnated glass microfibre 10 cm strips. Developed iTLC strips were scanned with a LabLogic miniscan TLC reader with positron (β^+^) or gamma (γ) detectors for ^68^Ga and ^67^Ga, respectively (data analysed with Laura 6 software), or by Cyclone Plus phosphor filmless autoradiography imager (analysed with Cyclone Plus 5.0 software). HPLC was implemented using an Agilent Eclipse XDB C_18_ 5 μm 4.6 × 150 mm reversed phase (RP) column and Agilent 1200 series HPLC with ultraviolet (UV, 220 nm) and radioactivity (Elysia raytest) detection and Gina Star^TM^ software version 5.8. Gamma counting of tissue samples was performed using an LKB Wallac 1282 Compugamma Gamma Counter. Mass spectrometry analysis was conducted with a Thermo Exactive HR mass spectrometer with electrospray ionisation (ESI), running Xcalibur version 2.2 software. Nuclear magnetic resonance (NMR) spectra were acquired on a Bruker Avance III HD NanoBay 400 MHz NMR spectrometer (Ascend^TM^ magnet) and analysed with Bruker Topspin 4.2 software. Inductively coupled plasma-mass spectrometry (ICP-MS) determination of gallium in tissues was conducted in the London Metallomics Facility at King’s College London using a PerkinElmer NexION 350D, with Syngistix version 1.0. LC/MS was acquired through an Agilent Eclipse XDB C_18_ 5 μm 4.6 × 150 mm RP column on an Agilent liquid chromatograph (1200 Series) with UV detection at 254 nm, connected to an Advion Expression LCMS mass spectrometer with electrospray ionisation source. Analysis was performed with Advion Mass Express software (version 6.4.16.1). Elemental microanalysis (for C, H, N) was performed by London Metropolitan University’s elemental analysis service.

### 5.2. Synthesis of KP46

The synthesis of KP46 was based on a method previously reported by Collery et al. [22]. In brief, 1.175 g (8 mmol) of 8-hydroxyquinoline, dissolved in 10% acetic acid (40 mL), was added to 0.559 g (2 mmol) of gallium (III) nitrate [Ga(NO_3_)_3_.H_2_O]. The mixture was heated under reflux to 80 °C for one hour before being filtered, and the residue was washed sequentially with hot water, cold water, and diethyl ether. The final yellow solid compound was dried overnight at 100 °C in a drying oven (yield 0.6232 g, 1.2 mmol, 60%).

### 5.3. Preparation of “Tracer” [^68^Ga]KP46, [^67^Ga]KP46, and [^68^Ga]Gallium Acetate

^68^Ga complex formation with 8-hydroxyquinoline was performed according to a modified method used by Yano et al. [56], in which 200 µL of a solution of 8-hydroxyquinoline in ethanol (1 mg/mL) was mixed with 200 µL of sodium acetate (0.5 M in sterile H_2_O, pH = 9) and 100 µL (8–10 MBq) of ^68^Ga from the most radioactive generator eluate fraction. This mixture contained 2.76 mM 8-hydroxyquinoline and approximately 0.98 nM ^68^Ga in a total volume of 0.5 mL at the time of preparation, with pH = 5.5–6.0. [^68^Ga]Ga-acetate was prepared similarly, but we replaced the ethanolic 8-hydroxyquinoline ligand solution with ethanol only. For in vivo studies, higher activity and more concentrated samples of [^68^Ga]KP46 and [^67^Ga]KP46 were required, which were produced and concentrated by the following scaled-up method: for ^68^Ga-labelling, 1 mL of the highest-activity ^68^Ga generator eluate fraction (typically 240–270 MBq) was added to a mixture of 2 mL of 8-hydroxyquinoline in solution in ethanol (1 mg/mL) and 1 mL of sodium acetate (0.5 M in sterile H_2_O, pH = 9). The final mixture pH was 5.5. In the case of [^67^Ga]KP46 radiolabelling, after ^67^Ga citrate to ^67^Ga chloride conversion (see Appendix A), 0.5 mL of ^67^Ga chloride solution was added to a mixture of 1 mL of freshly dissolved 8-hydroxyquinoline in ethanol (1 mg/mL) and 0.5 mL sodium acetate (0.5 M in sterile H_2_O, pH = 9). The final mixture pH was 5.5. The ^68^Ga and ^67^Ga mixtures were diluted (1:2 *v*/*v*) with sterile water and loaded into a Sep-Pak Light C-18 cartridge that had been preconditioned with 5 mL of ethanol and 5 mL of water. The radioactivity trapped on the cartridge was eluted with 500 µL of ethanol in 5 × 100 µL fractions (recovering >90% of the radioactivity in all cases), the most radioactive of which was diluted tenfold with phosphate-buffered saline. Radiochemical purity was evaluated by iTLC (CHCl_3_:CH_3_OH, 95:5%, *v*/*v*; [^68/67^Ga]KP46: Rf = 1; [^68^Ga]Ga-acetate: Rf = 0).

### 5.4. Preparation of [^68^Ga]KP46 and [^67^Ga]KP46 for In Vivo Oral Administration Studies

To prepare radiolabelled (^68^Ga and ^67^Ga) bulk KP46, the most active ethanolic fraction (40 μL) of [^68^Ga]KP46 and [^67^Ga]KP46 eluted from the C18 column (as described above for preparation of tracer-level solutions) was combined with a 20 mM KP46 solution in dimethylsulfoxide (20 μL) and a 0.4 g/mL solution of polyethylene glycol (PEG, to maintain solubility) in water (180 µL) and was made up to 400 μL with water (160 μL). The tracer-level samples for oral administration were prepared in the same manner but with 20 μL of KP46-free DMSO in place of the KP46 solution in DMSO. The samples were analysed by HPLC and radio-iTLC prior to use in animal studies to show that radiolabelled KP46 and KP46 behaved similarly.

### 5.5. Determination of the Distribution/Partition Coefficients (logD and logP) of [^68^Ga]KP46 and [^68^Ga]Gallium Acetate

Tracer-level [^68^Ga]KP46 and [^68^Ga]Ga-acetate solutions (5–20 µL, 0.3–0.5 MBq) were added to a pre-equilibrated mixture (500 µL/500 µL) of octanol/water (for log*P* measurement) or octanol/PBS (for log*D*_7.4_ measurement). Samples were vortexed for two minutes in a Multi Vortex Mixer V-32 (Grant Bio). From each layer, a sample of 200 µL was taken and its radioactivity measured using a gamma counter.

### 5.6. [^68/67^Ga]KP46 and [^68/67^Ga]Gallium Acetate Binding to Serum Proteins In Vitro

Solutions of human apo-transferrin (apo-Tf, 2 mg/mL) and human serum albumin (HSA, 50 mg/mL) (both purchased from Sigma Aldrich) were dissolved in aqueous NaHCO_3_ (5 mM, pH = 8) and PBS (pH = 7.4), respectively. An amount of 1 MBq of tracer-level [^68^Ga]KP46 (13–20 µL) or [^68^Ga]gallium acetate (20–25 µL) was added to 1.2 mL of the apo-Tf solution and to a similar apo-Tf-free NaHCO_3_ solution (control). Similarly, [^68^Ga]KP46 and [^68^Ga]gallium acetate solutions were added to 1.5 mL PBS containing HSA or HSA-free PBS (control). The samples were incubated for 60 min at 37 °C (pH = 7–7.5). From each mixture, 0.5 mL was loaded into PD MidiTrap G-25 size exclusion chromatography cartridges that were preconditioned with 8 mL of NaHCO_3_ (5 mM) (for apo-Tf binding) or PBS (for HSA binding). Fractions (0.5 mL) were collected and counted with a Wallac gamma counter. Protein absorbance at 280 nm was measured using a Nanodrop spectrophotometer. To investigate the speciation of the Ga–KP46 complex in the presence of apo-Tf at later time points (up to 48 h), the experiment was repeated with ^67^Ga.

### 5.7. Cellular Uptake of [^68^Ga]KP46 In Vitro

One million A375 (human melanoma) cells were suspended in 0.5 mL of Dulbecco’s Modified Eagle’s Medium (500 mL DMEM, low-glucose medium with 50 mL 10% FBS, 5 mL of L-glutamine, and 5 mL of penicillin and streptomycin) in 1.5 mL Eppendorf tubes. An amount of 10 µL of no-carrier-added [^68^Ga]KP46 or [^68^Ga]gallium acetate was added to the cell suspensions (giving a ^68^Ga concentration of ca. 20 pM) and to similar tubes containing medium but no cells (to check and compensate for adhesion of radioactivity to tubes). Samples were incubated for 60 min at 37 °C under 5% CO_2_ and then centrifuged (Sci Spin Mini) for three minutes at 1500 RPM. The supernatants were transferred to fresh Eppendorf tubes. The pellets were washed twice with 0.5 mL PBS, following the same centrifugation protocol, and washes were added to the supernatant tubes. Supernatants and cell pellets were measured using a gamma counter to determine the average activity in cell pellets as a % of the total activity for each sample. The intracellular-to-extracellular ^68^Ga concentration ratio was also estimated by correcting for the different volume of supernatant and cell pellet using an estimated spherical cell diameter of 17.5 μm, which leads to an estimated volume of the cell pellet (10^6^ cells) of 2.8 μL.

### 5.8. In Vivo Studies

All animal experiments were performed in accordance with the Animals (Scientific Procedures) Act, 1986, with protocols approved by the Animal Welfare and Ethical Review Body for King’s College London under project and personal licences approved by the UK Home Office. Tumour xenograft studies were performed on female NSG mice or female athymic nude mice after injecting them subcutaneously in the right shoulder with 2.5 × 10^6^ A375 cells in 100 µL of PBS. All in vivo studies were conducted between 14 and 24 days after inoculation, with tumour volumes between 100 mm^3^ and 580 mm^3^ (see Appendix A).

*Intravenous [^68^Ga]gallium acetate and [^68^Ga]KP46:* NSG mice bearing A375 xenografts were fasted for 12–14 h before the start of the experiment. Mice were anaesthetised with isoflurane and injected with tracer level [^68^Ga]gallium acetate (group A, n = 5) or [^68^Ga]KP46 (group B, n = 5) (13–18 MBq, <200 μL) via the tail vein. Mice were kept under anaesthesia during PET/CT imaging for 4 h (see protocol below), then culled by neck dislocation. The organs were harvested, weighed, and gamma counted. Additional similar studies with athymic nu/nu mice were conducted similarly.

*Oral [^68/67^Ga]KP46:* The protocol for oral administration experiments reported here was devised on the basis of pilot experiments described in Appendix A. For short-term (up to 4 h) studies with ^68^Ga, awake (nonanaesthetised, to avoid suppressive effects of anaesthesia on digestive tract activity) mice that had been fasted for up to 14 h were given [^68^Ga]KP46 (5–13 MBq) (group C, “tracer”, n = 3) or [^68^Ga]KP46 and KP46 (5–8.5 MBq) (group D, “bulk”, n = 3) by oral gavage. The dose given orally to group D animals, prepared as described above, was 0.4 μmol in 400 μL, molar activity ca. 25 MBq/μmol. The mice were fasted for three hours before being anaesthetised and PET/CT scanned dynamically for one hour (see below for scanning protocol). After scanning, mice in both groups were culled by neck dislocation while still anaesthetised four hours post-oral administration for organ harvesting and gamma counting. Samples of blood, urine and contents of stomach, small intestine and large intestine (the organs/tissue with measurable activity; tumours were not included as they were insufficiently radioactive) from the mice in group C were used for octanol extraction (shake-flask method) to separate water-soluble (released) and lipid-soluble (intact KP46) ^68^Ga. Serum samples were prepared by placing blood samples in a serum separator tube (brand VS367954, Fisher Scientific Ltd., Loughborough, UK) and centrifuging at 3000 rpm for 10 min. Serum (200–300 µL) and urine samples (150–180 µL, taken from the excised bladders) were directly added to Eppendorf tubes containing a pre-equilibrated mixture of octanol (500 µL) and water (500 µL). The tubes were vortexed for two minutes on a Multi Vortex Mixer V-32 (Grant Bio). After separation of layers, a sample of 400 µL from each layer was transferred to fresh Eppendorf tubes and analysed with a gamma counter. Stomach content (72–77 mg), small intestine content (35–88 mg), and large intestine content (86–100 mg) samples from each mouse were analysed similarly by adding them directly to Eppendorf tubes containing a pre-equilibrated mixture of octanol (500 µL) and water (500 µL) before vortexing and centrifuging at 10,000 rpm for 10 min at 4 °C. From each layer, 430 µL was transferred to fresh Eppendorf tubes and analysed with a gamma counter; the solid residues were also counted. Tissues from mice in group D were used after radioactive decay for measurements of nonradioactive ^69^Ga content.

For long-term (up to 48 h) studies, a similar protocol was followed, but SPECT imaging using ^67^Ga was repeated at intervals up to 48 h instead of PET imaging up to only 4 h postadministration. Athymic nude *nu*/*nu* mice bearing A375 xenografts were fasted for 7–14 h and divided in two groups: E (n = 4) and F (n = 4). Awake (nonanaesthetised) mice were orally administered with [^67^Ga]KP46 (2–11 MBq, “tracer”, group E) or with [^67^Ga]KP46 and KP46 (1–11 MBq, “bulk”, group F, given 0.4 μmol KP46 with molar activity ca. 15 MBq/μmol) and fasted for three hours. Mice in both groups were then anaesthetised and SPECT/CT scanned (see scanning protocol below) for one hour. Mice were then allowed to recover and feed. At 24 h postadministration, mice were reanaesthetised and rescanned, then group E mice were culled while still anaesthetised, and tissues were harvested for ex vivo gamma counting. Group F mice, which showed sufficient remaining radioactivity for further scanning at later time points, were allowed to recover and feed after the 24 h scan, and they were reanaesthetised and rescanned 48 h post-oral administration before culling and harvesting of organs for gamma counting and ^69^Ga measurement by ICP-MS.

### 5.9. Scanning Protocols

PET/CT images were acquired with a nanoScan^®^ PET/CT (Mediso Medical Imaging Systems, Budapest, Hungary) scanner operating in list mode using a 400–600 keV energy window and a coincidence window of 1:3. CT scans were acquired for anatomical reference and attenuation correction (55 keV X-ray, exposure time 1000 ms, and 360 projections and pitch 1). PET projection data were reconstructed using the Tera-tomo^®^ software package provided with the scanner —a Monte Carlo-based fully 3D iterative algorithm with four iterations, six subsets, and 0.4 mm isotropic voxel size; corrections for attenuation, scatter, and dead-time were enabled. The data were then visualised and quantified using VivoQuant^©^ v.2.50 (InviCro, Boston, MA, USA) software. SPECT/CT imaging was performed on a NanoSPECT/CT Silver Upgrade scanner (Mediso; 4 heads, 4 × 9 1.0 mm multipinhole collimators) in helical scanning mode using energy windows centred around 93.20 ± 20% keV (primary), 184.60 ± 20% keV (secondary), and 300.00 ± 20% keV (tertiary). CT images were acquired using 45 kVp tube voltage and 1000 ms exposure time in 180° projections. SPECT/CT data sets were reconstructed using the HiSPECT 1.4.2611 (SciVis, Gottingen, Germany) reconstruction software package using standard reconstruction with 35% smoothing and 9 iterations. Images were coregistered and analysed using VivoQuant v2.50 (InviCro, Needham, MA, USA).

## Figures and Tables

**Figure 2 molecules-28-07217-f002:**
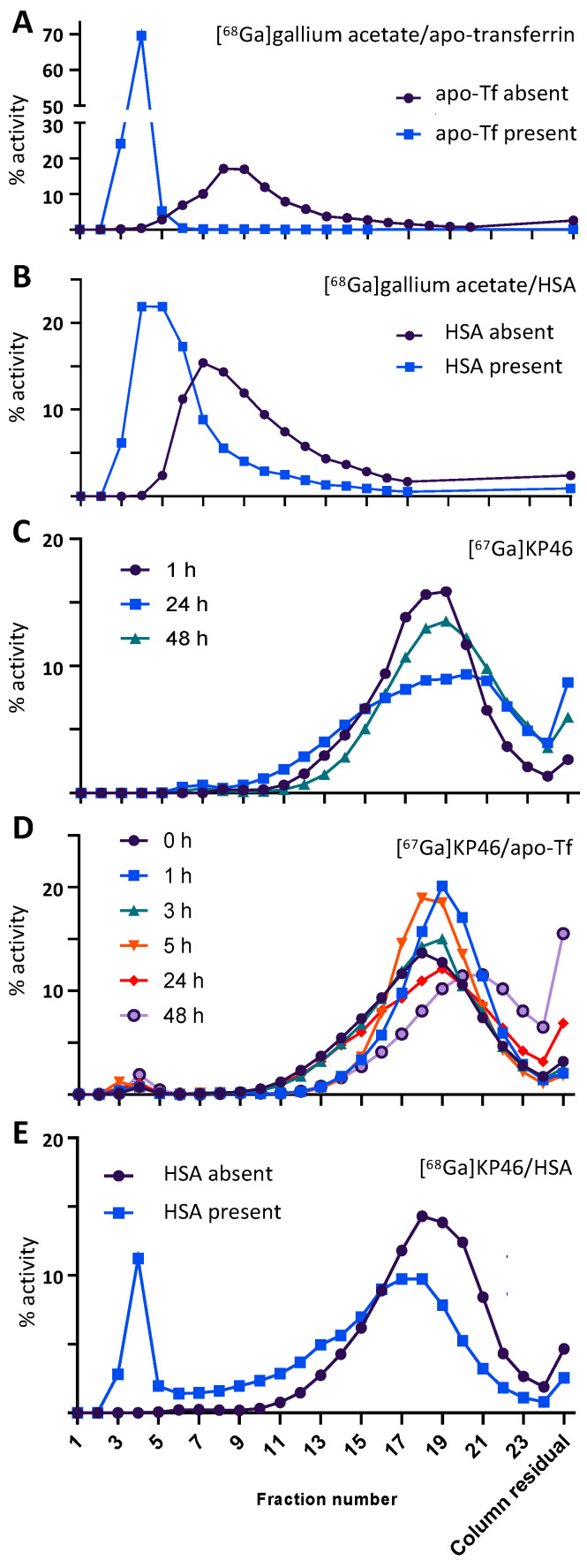
Size exclusion elution (PD MidiTrap G-25) profiles of [^68^Ga]gallium acetate and [^68/67^Ga]KP46 in the presence and absence of apo-transferrin (apo-Tf) and human serum albumin (HSA). Vertical axis represents % of total activity eluted. (**A**) [^68^Ga]gallium acetate incubated for 1 h with and without apo-Tf in the presence of bicarbonate, showing essentially quantitative binding of ^68^Ga to apo-Tf; (**B**) [^68^Ga]gallium acetate incubated for 1 h with and without HSA, showing partial binding of radioactivity to HSA; (**C**) [^67^Ga]KP46 incubated in bicarbonate buffer over a period of 48 h, showing minor changes in elution profile, used as a control for (**D**) [^67^Ga]KP46 incubated with apo-Tf in the presence of bicarbonate, sampled over a period of 48 h, showing very minor binding of ^67^Ga to apo-Tf at later time points; (**E**) [^68^Ga]KP46 incubated for 1 h with and without HSA, showing significant binding of radioactivity to HSA.

**Figure 3 molecules-28-07217-f003:**
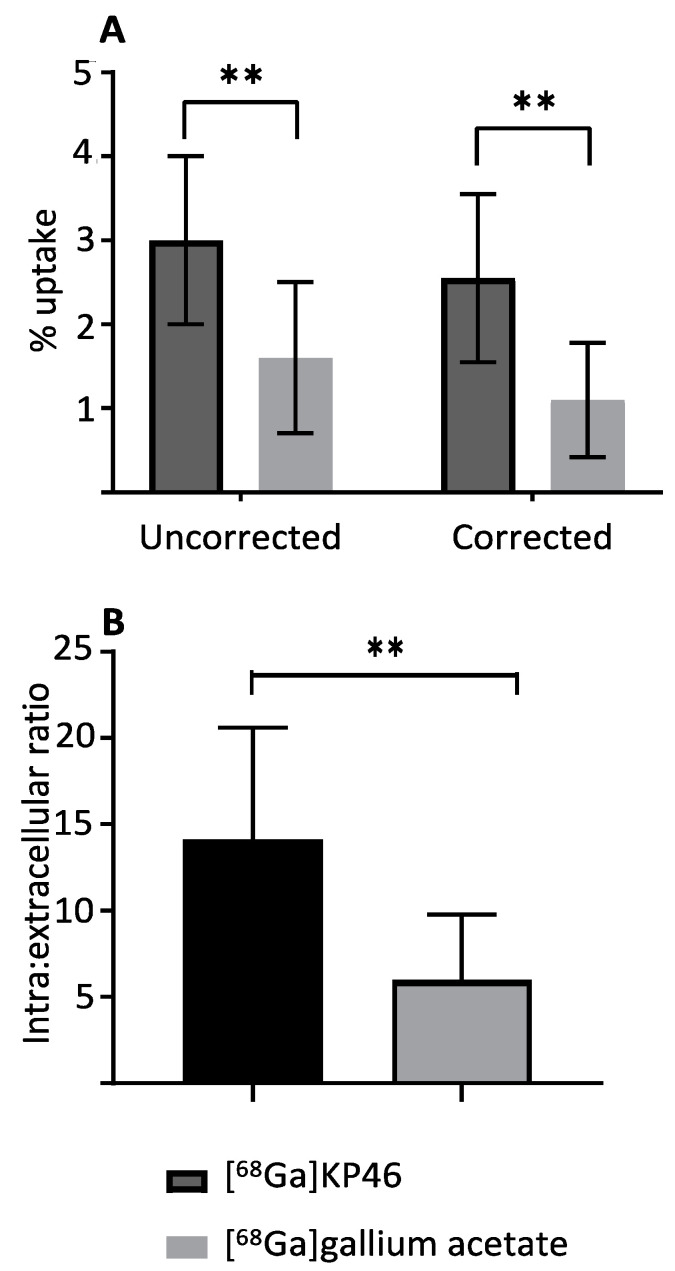
Uptake of [^68^Ga]KP46 and [^68^Ga]gallium acetate in cultured A375 cells after one hour of incubation in DMEM media containing FBS (n = 3 independent experiments). (**A**) Expressed as % radioactivity added, uncorrected (left) and corrected (right) for radioactivity binding to the plate in the absence of cells; (**B**) expressed as ratio of intracellular to extracellular ^68^Ga concentration (using corrected data). Uptake of [^68^Ga]KP46 was significantly higher (** *p* < 0.01, unpaired two-tailed *t*-test) than that of [^68^Ga]gallium acetate. Data are reported as mean ± SD.

**Figure 4 molecules-28-07217-f004:**
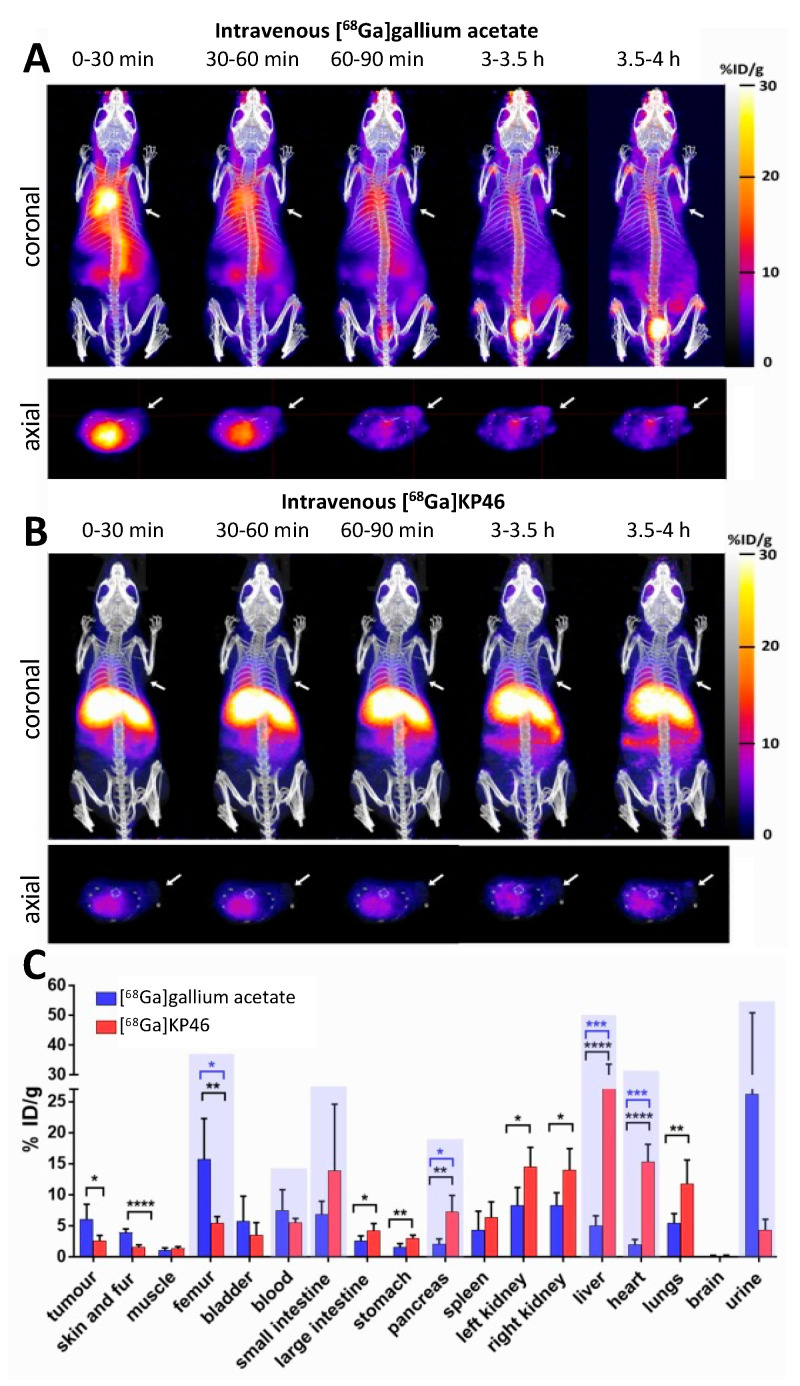
Biodistribution of ^68^Ga in NSG mice bearing A375 tumour xenografts on the right shoulder, during 4 h period after i.v. injection of [^68^Ga]gallium acetate (group A) and [^68^Ga]KP46 (group B). (**A**) Dynamic PET/CT images (maximum intensity projection) at different time points post-i.v. injection of [^68^Ga]gallium acetate; (**B**) dynamic PET/CT images (maximum intensity projection) at different time points post-i.v. injection of [^68^Ga]KP46. For both (**A**,**B**), the top panel shows dorsal aspect maximum intensity projections, and the bottom panel shows axial transverse sections. The white arrows show the position of the tumour. (**C**) Ex vivo biodistribution of [^68^Ga]Ga-acetate (group A, blue) and [^68^Ga]KP46 (group B, red) at 4 h post-i.v. injection. Values are expressed as mean ± SD (n = 5). Statistical analysis was performed for each organ using a *t*-test (black *). For the groups shaded with blue background, unequal variances were obtained (F test), and therefore Welch-adjusted *t*-test was also carried out (blue *). *p* values are defined as follows: * = *p* ≤ 0.05, ** = *p* ≤ 0.01, *** = *p* ≤ 0.001, **** = *p* ≤ 0.0001.

**Figure 5 molecules-28-07217-f005:**
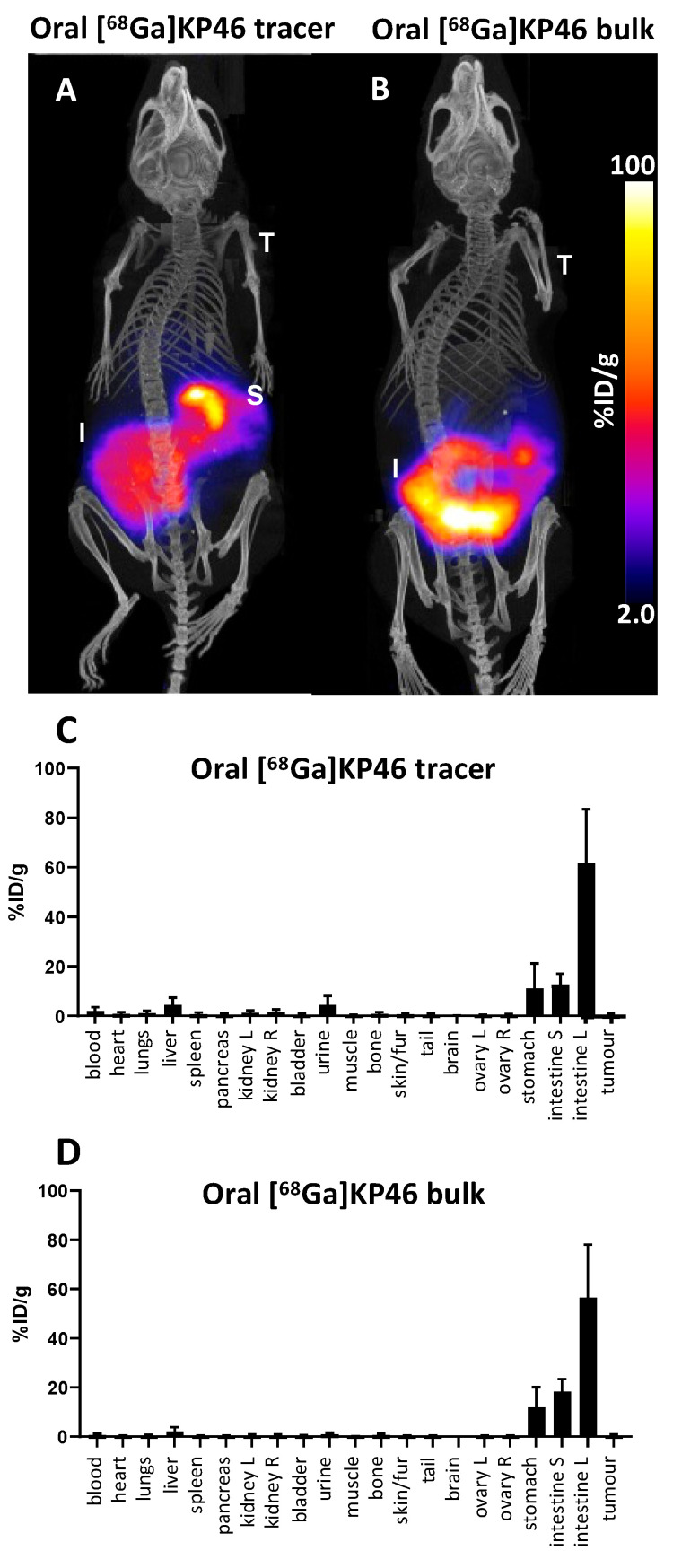
Biodistribution of ^68^Ga in *nu*/*nu* mice bearing A372 tumour xenograft 4 h after oral administration of [^68^Ga]KP46 (tracer level, group C; bulk level, group D), showing minimal absorption from gastrointestinal tract in both cases. (**A**) Exemplar PET/CT scans, group C; (**B**) exemplar PET/CT scan, group D. Both show PET/CT maximum intensity projections (T = tumour, S = stomach, I = intestines). (**C**,**D**) Ex vivo % injected dose/g (%ID/g) data for tracer (**C**) and bulk (**D**) administration, n = 3. Error bars show standard deviation.

**Figure 6 molecules-28-07217-f006:**
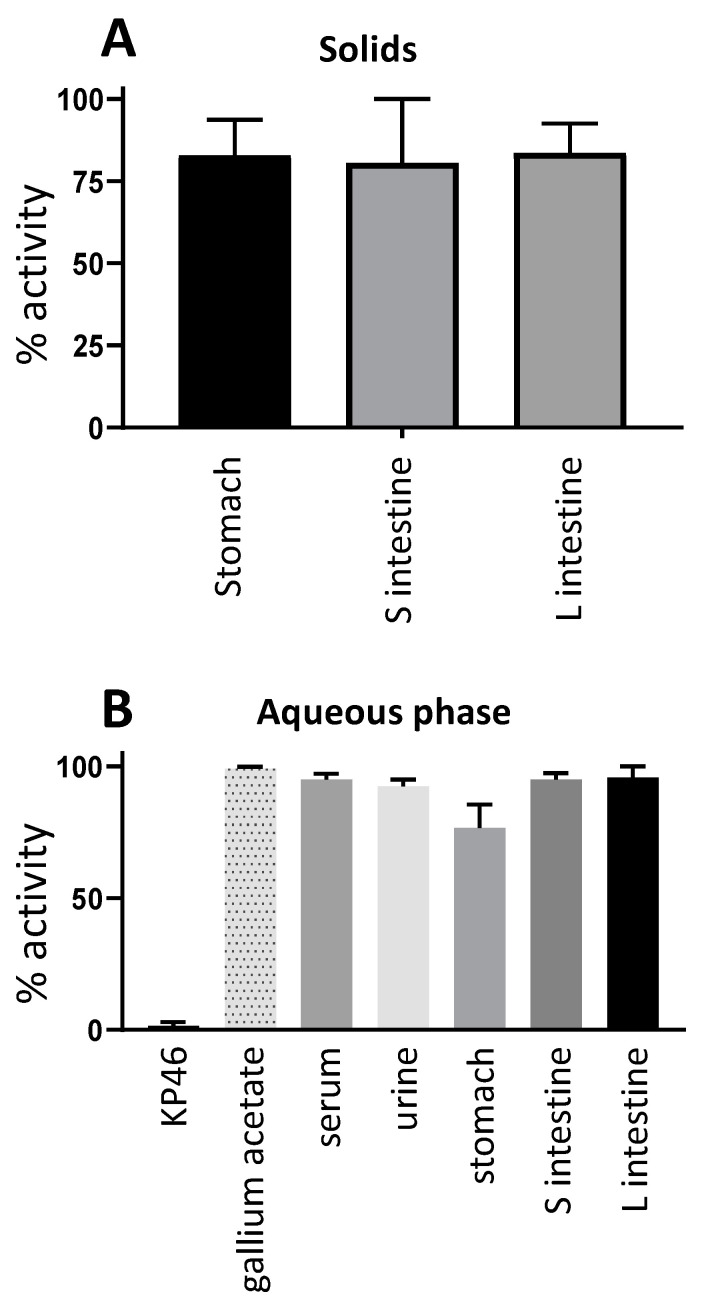
Solvent extraction of ^68^Ga from organs of mice given oral tracer dose of [^68^Ga]KP46 (group C), 4 h postadministration. (**A**) % radioactivity remaining in solids during extraction of stomach and intestine contents with octanol and water. (**B**) % of soluble radioactivity in aqueous phase during octanol extraction. Samples of [^68^Ga]KP46 and [^68^Ga]gallium acetate are included as controls. Values are expressed as mean ± SD.

**Figure 7 molecules-28-07217-f007:**
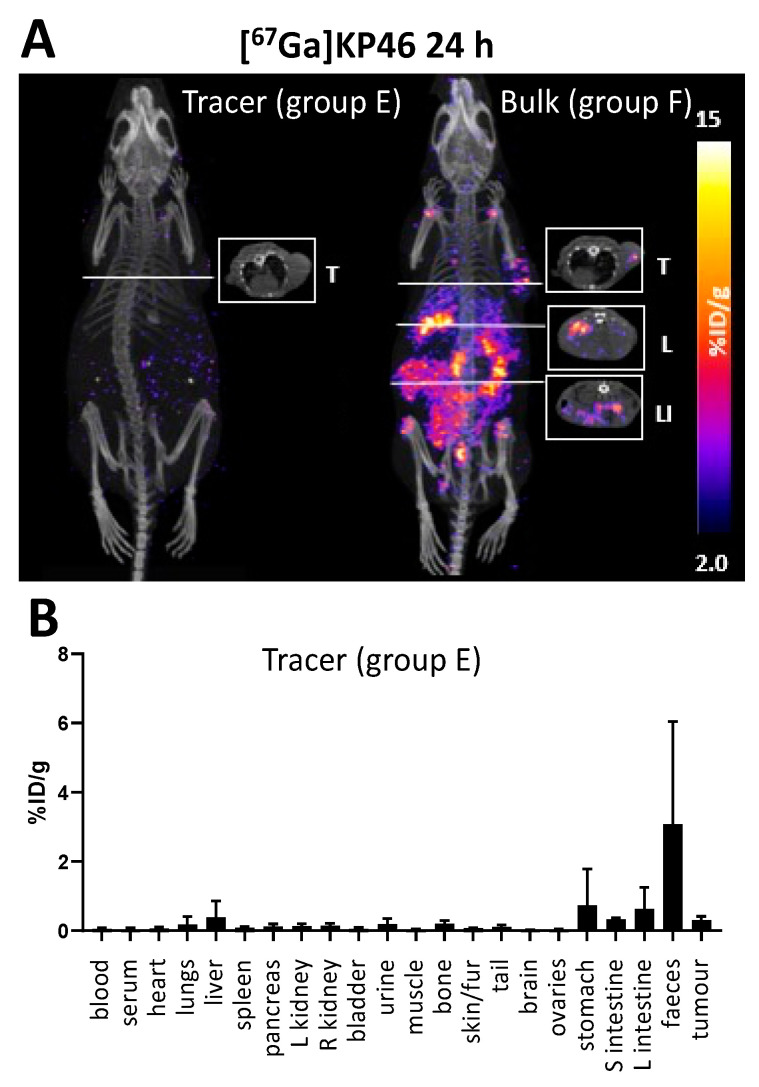
Biodistribution of ^67^Ga in nu-nu mice with A375 tumour xenograft, 24 h after oral administration of tracer level (group E) or bulk level (group F) [^67^Ga]KP46. (**A**) SPECT/CT scans, reconstructed as coronal maximum intensity projections with (inset) axial sections. T = tumour, L = liver, S = stomach, LI = large intestine. Left image shows that radioactivity had been largely eliminated from the bodies of group E mice by 24 h, and hence, mice in this group were sacrificed for ex vivo biodistribution rather than reimaged at 48 h. (**B**) Ex vivo biodistribution 24 h after tracer-level oral administration (group E), expressed as mean %ID/g. Error bars represent standard deviation (n = 4).

**Figure 8 molecules-28-07217-f008:**
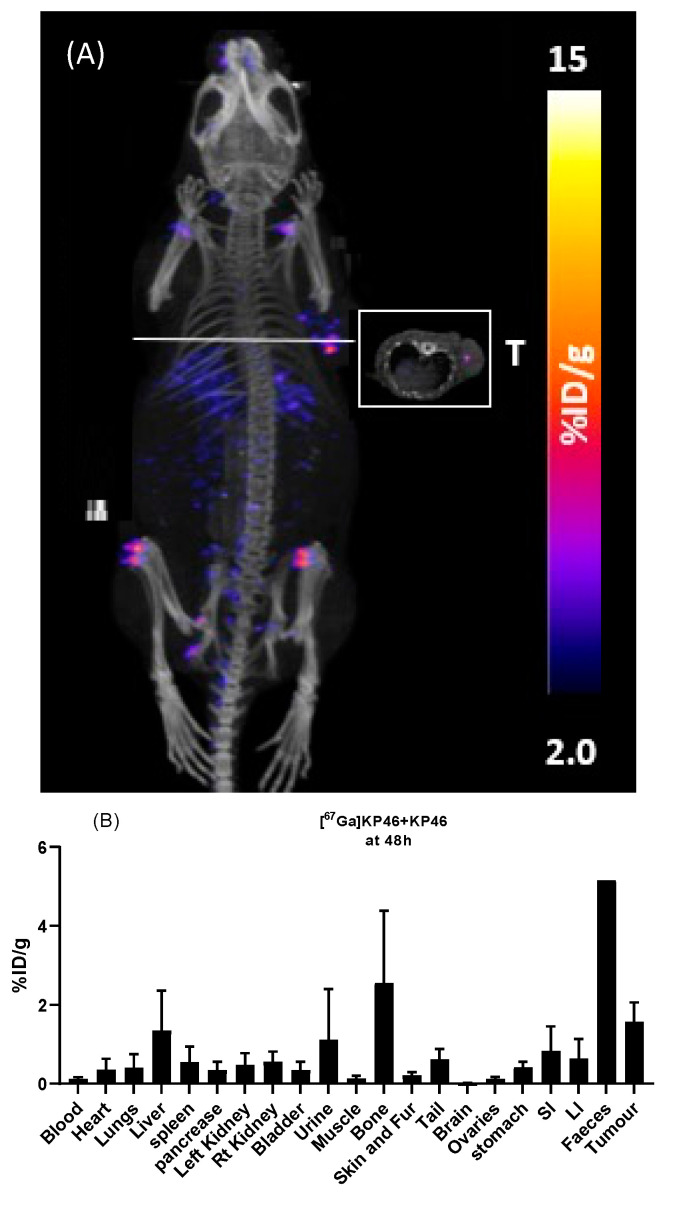
Biodistribution of ^67^Ga in nu-nu mice with A375 tumour xenograft, 48 h after oral administration of bulk level (group F) [^67^Ga]KP46. (**A**) SPECT/CT scans, reconstructed as maximum intensity projections with (inset) axial section. T = tumour. (**B**) Ex vivo biodistribution 48 h after tracer-level oral administration (group F), expressed as mean %ID/g. Error bars represent standard deviation (n = 4 except faeces, where n = 1).

## Data Availability

All data are provided within the published article or Appendix A. Raw imaging data underpinning in this study are available on request from the corresponding author.

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
