# Peer review of "In Vivo Trafficking of the Anticancer Drug Tris(8-Quinolinolato) Gallium (III) (KP46) by Gallium-68/67 PET/SPECT Imaging"

_molecules, 2023, doi:10.3390/molecules28207217_

Round 1
Reviewer 1 Report
The manuscript entitled “In vivo trafficking of the anti-cancer drug tris(8-quinolinolato) gallium (III) (KP46) by gallium-68/67 PET/SPECT imaging” describes the preparation and analysis of radioactive KP46 analogues (with 68Ga and 67Ga). Additionally, in vitro and in vivo studies using the radioactive agents were performed to clarify the pharmacokinetics and the gallium trafficking/delivery following the administration (i.v. and oral) of the KP46 complex. The authors’ thorough effort can be published after implementing the following minor corrections:
1. The whole manuscript is understandable, but many sections/sentences are too long, complicated, convoluted, repetitive and verbose. Understandably, each author has different writing style. However, the manuscript would definitely benefit from concision. The “Discussion” section should be shortened. Also, a “Conclusions” section has not been properly defined (lack of the title).
2. Page 3 line 112: the authors introduce the subject of RCY but all RCY values are missing. Please add the RCY values for each agent together with the molar activities.
3. Size exclusion elution of radioactive agents with Apo-Tf and HAS: the authors should indicate the recovery of radioactivity from the column (for each agent). There is a short mention of variable recoveries in page 5 (lines 155-157) but no values have been given.
4. The authors should add one line to explain why they plan to scan the mice 4 h post radioactive agent administration (and not 1, 2 or 3 h).
5. The authors mention “tumour-to-background ratios” few times throughout the manuscript but no values have been provided. The authors should add tables with those ratios (for each in vivo experiment, in Supporting Info) to corroborate their statements.
6. For completeness and clarity (to spare the reader from guessing the uptake values from the graphs), the authors should provide tables with the biodistribution values (in Supporting Info) for each in vivo study.
7. Page 7, line 193: The authors state that the tumor uptake post 68Ga-acetate injection is “visible” on PET images. Sorry to disagree, but it is very difficult to identify the tumor in figure 4A. Maybe because of the small size of the images or of the low uptake and high background noise. Anyway, the term “visible” is a bit of a stretch since the tumor would not be easily identified in the absence of the white arrows.
8. Page 7, line 201-202 and Fig S8: It is not clear why the mentioned study (i.e., biodistribution of [68Ga]KP46 2h p.i.) is relevant to the story. Please elaborate (briefly).
9. Page 10, line 289-290: The authors mention results from the octanol extraction for group E but a graph with the values is missing (as done for group D and Figure S9, and group F and Figure S10).
10. Why do the authors use two different mice models (athymic nudes and NSG)?
11. Material and methods: Why are the figures cited in this section? Usually figures are cited in the “Results” section (where the figures are usually located). It is confusing.
Page 16, line 575, Supporting Info, page 4: The authors mention elemental analysis, but the results are not shown anywhere. Please add.
Page 18, line 639: please indicate the concentration of the radioactive agents added to the cells.
Page 18, line 671 and Page 19, line 700: Please indicate the volume of the dose given by oral gavage.
12. Some aesthetic remarks: Misaligned legend in Figure 4C and incomplete cropping of Figure 8A (top left corner). Page 9, line 260: it’s Figure S11, not S4.
13. Supporting Info: There is a general lack of attention to details.
The figures are shown in a weird order. The numbering is not following an incremental succession based on the citation in the main manuscript. It is confusing and unpleasant.
Figure S3 (Top): Some of the labels on the top mass spectrum are wrong (e.g., Ga2(8HQ)2(O2CH)+ should be Ga2(8HQ)4(O2CH)+);
Figure S5: Unfortunately, the non-radioactive strip photos are not identifiable in the version of the document supplied by the editor (there is just a long blue band with a yellow arrow pointing nothing recognizable). Please double check that the photos will be clearly visible in the published version.
Figure S6: The addition of a table with the logD/P values would be useful to spare the reader from guessing the numbers from the graph.
Page 10, end of top section: It’s Fig S8, not S9.
Page 10: “Developing a method for the oral administration…” Please indicate the volume of the orally administered radioactive agent solution.
Figure S13C: Mismatched numbers of X-axis ticks (19 in total) and organ/tissues (21 in total). Please amend.
Generally, the English is good (only a couple of missing/extra articles/prepositions). However, many sections/sentences are long, convoluted, and unnecessarily wordy. The flow of the manuscript would benefit from some simplification.
Author Response
Revierwer's comments in plain text; authors' responses highlighted yellow.
The whole manuscript is understandable, but many sections/sentences are too long, complicated, convoluted, repetitive and verbose. Understandably, each author has different writing style. However, the manuscript would definitely benefit from concision. Over long sentences have been broken down into shorter ones throughout. The “Discussion” section should be shortened. Having found no significant tracts of text that are not essential to the discussion for readers’ benefit, and in the absence of restrictions laid down by the journal, the authors prefer not to remove significant sections of the Discussion. Also, a “Conclusions” section has not been properly defined (lack of the title). A Conclusion heading has now been inserted at the end of the Discussion, to clearly define the existing conclusion paragraph.
- Page 3 line 112: the authors introduce the subject of RCY but all RCY values are missing. Please add the RCY values for each agent The required information has been added to section 5.3 for n.c.a. [67Ga]KP46 and [68Ga]KP46; the concept of RCY for the [68Ga]gallium acetate has no meaning since no separation or purification was carried out, all radioactivity remained in the sample. Use of the term “yield” in these circumstances in our manuscript is inappropriate and has now been removed. The same applies to the “bulk” carrier added samples, which were prepared by addition of the purified tracer complex to the bulk drug, without any subsequent separation; hence these yields are 100% but this is not a meaningful number, together with the molar activities. Molar activity: for “tracer level” [68/67Ga]KP46 and [68Ga]gallium acetate, the samples were “no carrier added” and molar activity was not measurable. Calculated estimates of the gallium concentrations are given in the text (picomolar to nanomolar range). For [67/68Ga]KP46 prepared in bulk, the final KP46 concentration was 1.53 mM after dilution, in a volume of 400 uL administered, total amount administered is 0.4 umol, activity 10 MBq; 1 umol = 2.5x10 = 25MBq/umol (now stated p 19)
- Size exclusion elution of radioactive agents with Apo-Tf and HAS: the authors should indicate the recovery of radioactivity from the column (for each agent). There is a short mention of variable recoveries in page 5 (lines 155-157) but no values have been given. This information is provided in the charts in Fig. 2. A reference to this has been added to p5.
- The authors should add one line to explain why they plan to scan the mice 4 h post radioactive agent administration (and not 1, 2 or 3 h). The explanation for this is provided already, p. 7 line 214ff. This is only relevant for orally administered tracer. I.v.-administered animals were imaged dynamically from 0-4 h.
- The authors mention “tumour-to-background ratios” few times throughout the manuscript but no values have been provided. The authors should add tables with those ratios (for each in vivo experiment, in Supporting Info) to corroborate their statements. Thank you for drawing attention to this. The phrase “tumour-to-background” in fact occurs only once in the manuscript, and is used inappropriately as it implies quantitative measurement of this ratio, which is not the case. The ratios are not a useful element of the data or descriptor of biodistribution in this work. Tables of tumour-to-background ratios would be of no value – what would be chosen as “background”? Instead we have removed the term and made the point with alternative wording (p7)
- For completeness and clarity (to spare the reader from guessing the uptake values from the graphs), the authors should provide tables with the biodistribution values (in Supporting Info) for each in vivo study. Thank you for this suggestion. A table of all the values and standard deviations has been added (Table S1).
- Page 7, line 193: The authors state that the tumor uptake post 68Ga-acetate injection is “visible” on PET images. Sorry to disagree, but it is very difficult to identify the tumor in figure 4A. Maybe because of the small size of the images or of the low uptake and high background noise. Anyway, the term “visible” is a bit of a stretch since the tumor would not be easily identified in the absence of the white arrows. Indeed we do disagree. In Fig 4A the tumour uptake is clearly and objectively visible by 3-4 h, although not in the first 90 min. The text has been modified to specify that uptake is visible by 4 h.
- Page 7, line 201-202 and Fig S8: It is not clear why the mentioned study (i.e., biodistribution of [68Ga]KP46 2h p.i.) is relevant to the story. Please elaborate (briefly). These data corroborate the evidence presented in Fig. 4; because they do not provide time-matched comparison with the [68Ga]gallium acetate biodistribution, they do not fit well in Fig. 4 and this is why they are provided in the supplementary material and not the main text.
- Page 10, line 289-290: The authors mention results from the octanol extraction for group E but a graph with the values is missing (as done for group D and Figure S9, and group F and Figure S10). In this analysis only two values were recorded – serum and urine – and those are reported in the text; it seems like a waste of space to graphically present, or tabulate, two values.
- Why do the authors use two different mice models (athymic nudes and NSG)? There is no planned reason for this – the experiments were conducted over a long period during which interruptions in supplies during the COVID pandemic were encountered. Since there is no specific reason to choose one or the other, and the general biodistribution patterns are similar in both strains, we felt it appropriate not to make a complicating issue of it by discussing in the text. We remain of that view.
- Material and methods: Why are the figures cited in this section? Usually figures are cited in the “Results” section (where the figures are usually located). It is confusing. All but essential references to supplementary materials/figures have been removed.
Page 16, line 575, Supporting Info, page 4: The authors mention elemental analysis, but the results are not shown anywhere. Please add. Apologies for the omission, data have been added to supplementary information.
Page 18, line 639: please indicate the concentration of the radioactive agents added to the cells. done
Page 18, line 671 and Page 19, line 700: Please indicate the volume of the dose given by oral gavage. Done
- Some aesthetic remarks: Misaligned legend in Figure 4C Cannot find this error, but any error will be dealt with by providing a new figure and incomplete cropping of Figure 8A (top left corner).Again no error found, this will be dealt with by providing a new figure Page 9, line 260: it’s Figure S11, not S4. Thank you for spotting this, all references to supplementary figures have been rechecked and corrected.
- Supporting Info: There is a general lack of attention to details.
The figures are shown in a weird order. The numbering is not following an incremental succession based on the citation in the main manuscript. It is confusing and unpleasant. All references to supplementary figures have now been checked and corrected.
Figure S3 (Top): Some of the labels on the top mass spectrum are wrong (e.g., Ga2(8HQ)2(O2CH)+ should be Ga2(8HQ)4(O2CH)+); thank you for spotting these, they have been corrected
Figure S5: Unfortunately, the non-radioactive strip photos are not identifiable in the version of the document supplied by the editor (there is just a long blue band with a yellow arrow pointing nothing recognizable). Please double check that the photos will be clearly visible in the published version. Thank you, yes this it is a challenge to ensure these photos show the spots as clearly as in the real objects. We believe the spots are visible on close scrutiny but it is difficult in some cases.
Figure S6: The addition of a table with the logD/P values would be useful to spare the reader from guessing the numbers from the graph. Thank you for the suggestion, the values have been added to the Figure (they were also given in the text in the main manuscript).
Page 10, end of top section: It’s Fig S8, not S9. All references to supplementary figures have now been checked and corrected.
Page 10: “Developing a method for the oral administration…” Please indicate the volume of the orally administered radioactive agent solution. done
Figure S13C: Mismatched numbers of X-axis ticks (19 in total) and organ/tissues (21 in total). Please amend. Thank you for spotting this, corrected.
Comments on the Quality of English Language
Generally, the English is good (only a couple of missing/extra articles/prepositions). However, many sections/sentences are long, convoluted, and unnecessarily wordy. The flow of the manuscript would benefit from some simplification. Longer sentences have been broken up into more easily readble sentences.
Reviewer 2 Report
The work of Darwesh and co-authors focuses on the radiolabelling of KP46 with gallium 68 and 67 to study the drug's fate in vivo and in vitro. They reported properly the results and the discussion was fine. A few notes here:
Put the Y axis in Figure 2 (not just the caption).
In Figure 3 instead of %uptake would have been more reliable the use of nmol/cell like this the comparison with other publications can be easier.
I would delete/change from lines 524 to 528 since the translation of a tracer from mice to human is not only a matter or radionuclide and technologies availability but it's something deeper spanning from the dosimetry and chemistry till the clinical validation.
Author Response
Reviwer comments in plain text; authors' response highlighted yellow.
Put the Y axis in Figure 2 (not just the caption).Done
In Figure 3 instead of %uptake would have been more reliable the use of nmol/cell like this the comparison with other publications can be easier.Since these experiments were conducted with no-carrier-added tracer, comparisons of nmol/cell with other publications studying these drugs is not directly possible (since the concentration used here is tens of picomolar, much lower than experiments that use non-radioactive methods). We remain convinced that the measures used in Fig. 3 are the most informative for comparison of these two tracers.
I would delete/change from lines 524 to 528 since the translation of a tracer from mice to human is not only a matter or radionuclide and technologies availability but it's something deeper spanning from the dosimetry and chemistry till the clinical validation. We contend that this is an important point of information for readers involved in the field of gallium drugs, who may not be familiar with PET and SPECT clinical imaging, and we prefer to make it in the manuscript. However, we recognise the importance of the reviewer’s point and have therefore added the contingency that such human imaging experiments would be subject to appropriate dosimetry assessments and regulatory approvals.